# Improving the Performance and Stability of TIC and ICE

**DOI:** 10.3390/e25030512

**Published:** 2023-03-16

**Authors:** Tyler Ward

**Affiliations:** Department of Financial Engineering, NYU Tandon School of Engineering, 6 MetroTech Center, Brooklyn, NY 11201, USA; tw623@nyu.edu

**Keywords:** generalization error, overfitting, information criteria, entropy, AIC, TIC, ICE

## Abstract

Takeuchi’s Information Criterion (TIC) was introduced as a generalization of Akaike’s Information Criterion (AIC) in 1976. Though TIC avoids many of AIC’s strict requirements and assumptions, it is only rarely used. One of the reasons for this is that the trace term introduced in TIC is numerically unstable and computationally expensive to compute. An extension of TIC called ICE was published in 2021, which allows this trace term to be used for model fitting (where it was primarily compared to L2 regularization) instead of just model selection. That paper also examined numerically stable and computationally efficient approximations that could be applied to TIC or ICE, but these approximations were only examined on small synthetic models. This paper applies and extends these approximations to larger models on real datasets for both TIC and ICE. This work shows the practical models may use TIC and ICE in a numerically stable way to achieve superior results at a reasonable computational cost.

## 1. Preliminaries

The ICE methodology that is analyzed by this paper is described in detail in [1]. That paper also contains a great deal of introductory information regarding information criteria and generalization errors. This section contains a brief review to introduce some common notation needed for the topics of this paper.

Consider the model $g(x_{i}|\theta )$ that assigns a probability to the regressors $x_{i}$, and is parameterized by the parameters θ. Suppose the actual probability of $x_{i}$ is $f(x_{i})$. Suppose $X_{n}$ is a random variable corresponding to a sample of size *n* drawn from *f*, composed of $x_{i}$ for 0<i≤n. If the sample size is not specified (or understood to be 1), then we may write *X* instead. The usual definitions for this sort of analysis are listed below.

**Definition** **1**(Log Likelihood of θ over *X*)**.**
(1)$$\ell \left(\theta \right):=\frac{1}{n}\sum_{i=1}^{n}\log g\left(x_{i}|\theta \right)$$

**Definition** **2**(Expected Log Likelihood of θ over *f*)**.**
(2)$$L\left(\theta \right):=E_{x}\left[\log g\left(x|\theta \right)\right]$$
*where the expectation is taken over the distribution f(x).*

**Definition** **3**(Maximum Likelihood Estimate $\hat{\theta }$ of θ)**.**
(3)$$\hat{\theta }:=\underset{\theta \in \Theta }{\mathrm{argmax}}\ell \left(\theta \right).$$

**Definition** **4**(Negative Expected Hessian Matrix)**.**
(4)$$J\left(\theta \right):=-E_{X}\left[\partial_{\theta }^{2}\log g\left(X|\theta \right)\right]=-\int f\left(x\right)\partial_{\theta }^{2}\log g\left(x|\theta \right)dx,$$

**Definition** **5**(Fisher Information Matrix)**.**
(5)$$I\left(\theta \right):=E_{X}\left[\partial_{\theta }\log g\left(X|\theta \right)\partial_{\theta^{T}}\log g\left(x|\theta \right)\right].$$

For both I(θ) and J(θ), we define $\hat{J}\left(\theta ,X_{n}\right)$ and $\hat{I}\left(\theta ,X_{n}\right)$ to be their estimators computed from the dataset $X_{n}$. We may write $\hat{J}$ and $\hat{I}$ for simplicity when the meaning is clear.

### 1.1. Information Criteria and Generalization Error

A well known result by Stone [2] shows that the MLE is a biased estimator of the minimum KL-divergence:(6)$$E_{X_{n}}\left[-\ell \left(\hat{\theta }\left(X_{n}\right),X_{n}\right)\right]<E_{X_{n}}\left[-\ell \left(\theta_{0},X_{n}\right)\right],$$
because it is evaluated on the data $X_{n}$, which was used to fit $\hat{\theta }$. Cross-validation was developed as a model selection technique to select a model from a group that actually minimizes $E_{X_{n}}\left[\rho_{KL}\left(g_{\theta_{0}},g_{\hat{\theta }\left(X_{n}\right)}\right)\right]$ and not merely $E_{X_{n}}\left[-\ell \left(\hat{\theta }\left(X_{n}\right),X_{n}\right)\right]$ in the limit of large *n*. Takeuchi [3] and Akaike [4] explicitly modeled this bias (generalization error) of an estimation procedure $\theta (X_{n})$.

**Definition** **6**(Generalization Error of estimation procedure $\theta (X_{n})$)**.**
(7)$$b\left(\theta \left(X_{n}\right)\right):=E_{X_{n}}\left[\ell \left(\theta \left(X_{n}\right),X_{n}\right)-E_{X_{n}^{\prime }}\left[\ell \left(\theta \left(X_{n}\right),X_{n}^{\prime }\right)\right]\right].$$

Akaike’s Information Criterion (AIC) [4] was one of the earliest attempts to correct for this bias. AIC is able to correct for generalization errors when comparing MLE estimates for a restricted class of models. This work was extended by Takeuchi’s TIC [3] to expand the class of models, while still requiring that the MLE estimates be used for comparison.

In particular, note that it has long been known (e.g., in [4]) that for the MLE estimate $\hat{\theta }$ of a model with *m* parameters, the bias *b* is asymptotically $O(\frac{m}{n})$ almost surely. So, for instance,
(8)$$-\ell \left(\hat{\theta }\left(X_{n}\right),X_{n}\right)=-L\left(\hat{\theta }\left(X_{n}\right)\right)+O\left(\frac{m}{n}\right)$$
almost surely. Hence, for the MLE estimate $\hat{\theta }$ of a model with *m* parameters, we have that
(9)$$b\left(\hat{\theta }\left(X_{n}\right)\right)=O\left(\frac{m}{n}\right)$$
almost surely. Proofs of this fact are found in both [4], for a restricted subset of models, and [3] for a broader class of models.

The goal of AIC, TIC, and ICE is to reduce the generalization error by reducing the order of the $O(\frac{m}{n})$ term, by incorporating a more negative power of *n*. This does not guarantee superior performance. In the case of TIC particularly, numerical instability can cause this term to have an unexpectedly large constant factor. However, if numerical instability is effectively controlled, it is expected that many problems could benefit from these techniques for moderate sample sizes, as will be shown in later sections.

### 1.2. TIC

In [3], Takeuchi developed the information criterion
(10)$$TIC=-\ell \left(\hat{\theta }\right)+\frac{1}{n}tr\left(\hat{I}{\hat{J}}^{-1}\right).$$

The second term here may be periodically referred to as the “trace term,” as it appears in ICE as well in later sections. This was an extension of AIC, which had previously been developed by Akaike in [4]:(11)$$AIC=-\ell \left(\hat{\theta }\right)+\frac{m}{n}.$$

Here, for convenience, we use the convention that TIC (and AIC) is O(1). In other work, it is often multiplied by *n* to produce a result that is O(n).

Takeuchi then showed that AIC is a limiting case of TIC. It was shown by Stone in [2] that AIC model selection and model selection via cross validation are equivalent whenever AIC is valid. By extension, TIC is also equivalent to cross validation under these circumstances.

If two models are to be compared using TIC or AIC, then the model with the lower value of TIC is on average the better model. Given two models, $g_{1}$ with $TIC_{1}$ and $g_{2}$ with $TIC_{2}$, the model $g_{1}$ is actually the better model with probability
(12)$$p\left(\rho_{KL}\left(f,g_{1}\right)<\rho_{KL}\left(f,g_{2}\right)\right)=\frac{e^{-n*TIC_{1}}}{e^{-n*TIC_{1}}+e^{-n*TIC_{2}}}.$$
where $\rho_{KL}\left(f,g_{1}\right)$ is the KL divergence between the true distribution *f* and a model generated distribution *g*.

This follows directly from the fact that the exponential of a TIC value is a likelihood ratio, and the logic then proceeds in the usual way for likelihood ratio statistics [5].

In this way, TIC (as with any information criterion) can be used to select the better model from a family of fit models. However, it requires that all models be fit using maximum likelihood estimation (MLE).

### 1.3. Additional Information Criteria

Modern machine learning models often have a very large number of parameters, in some cases having more parameters than observations in the fitting set. Recalling Equation (Equation 9), it can be seen that using the MLE estimate $\hat{\theta }$ is likely to produce models that generalize poorly. For these models, information criteria have therefore fallen out of favor. Using an information criterion to choose the best model from a small set of models fit using MLE is unlikely to find an accurate model. If the number of models is very large, then Equation (Equation 12) dictates that the information criterion differences must be very large in order to reliably find the best model, and again the result is unlikely to perform well. Additionally, each fit of a model such as this may carry considerable expense, so producing a large number of fits to filter with information criteria may be cost prohibitive as well.

Konishi and Kitagawa [6] developed GIC, which extended TIC to no longer require MLE estimation, allowing regularization and similar generalization error reducing approaches. See [7] for an overview of typical regularization techniques that might be paired with GIC in this way. Unfortunately, GIC is not viable as written for modern machine learning models as it still has a form similar to Equation (Equation 10), and as discussed in [1,8,9], these equations are numerically unstable for large *m* (roughly m>20), regardless of *n*.

Ichiguro et al. developed an alternate approach named Extended Information Criterion (EIC) in [10]. The main idea is that TIC and AIC use a Taylor expansion of the generalization error, and then their correction terms are simply the leading order terms of that expansion. However, the generalization error itself (Equation (Equation 7)) actually takes the form of an expectation over the true distribution. This expectation may be computed directly over the empirical distribution, avoiding the need for an expansion.

Additional analysis of EIC was performed by Kitagawa and Konishi in [11]. Their analysis indicates that EIC (in its most basic implementation) adds noise to the objective function proportional to the sample size. This means it may not be appropriate for large datasets without adjustment, and adjustments to reduce this issue are then proposed and analyzed.

### 1.4. ICE

In the discussions of information criteria in the previous sections the models would be fit using MLE, or some other procedure, and then model selection would be performed afterwards using an information criterion. The exact fitting procedure is not specified. These approaches assume that some procedure can be found which will produce models with reasonable levels of accuracy, but that is hardly a given if the model has a very high parameter count *m* relative to the observation count *n*.

Though GIC allows the use of regularization (and various other techniques), $L_{2}$ regularization itself is not always effective. For instance, see Figures 1 and 3 from Section 4 of [1] for examples where $L_{2}$ regularization is not helpful. Models as simple as estimating mean and variance of a Gaussian through MLE are always harmed by $L_{2}$. This gives good cause to believe that cases where $L_{2}$ is not helpful, or not efficient, are fairly common. Approaches beyond regularization, such as early stopping or drop-out, tend to have hyper-parameters which can be difficult to estimate, just as regularization does. Additionally, there is little theoretical reason to believe that these approaches are reducing a generalization error efficiently.

An example of a highly parameterized model is a modern MNIST challenge leader [12] that has 1,514,187 parameters, but was fit on a dataset with only 30,000 observations. A discussion of why this often occurs within the field of machine learning is beyond the scope of this paper, but it is enough to know that this is an important use case for model fitting that is not well served by existing information criteria.

In [1], the ICE objective function is defined.

**Definition** **7**(ICE Objective)**.**
(13)$$-\ell^{*}\left(\theta \right)=-\ell \left(\theta \right)+\frac{1}{n}tr\left(I_{\theta }J_{\theta }^{-1}\right),$$
*Let $\theta^{*}$ denote the minimizer of* (Equation 13).

This takes the same form as TIC, but it was shown that with only slightly stronger assumptions (see Theorem 1 below) the trace term from Definition 7 is still the leading order generalization error term in a neighborhood around the MLE $\hat{\theta }$.

The important properties of this objective function are encapsulated in Theorem 1.

**Theorem** **1** (ICE Behavior). *Suppose the following conditions hold:*
*1.* 
*M satisfies White’s regularity conditions A1–A6 (see [13]).*
*2.* 
*$\theta_{0}$ is a global minimum of −L(θ) in the compact space Θ defined in A2.*
*3.* 
*There exists a ε>0 such that $-L\left(\theta_{0}\right)<-L\left(\theta_{1}\right)-\varepsilon$ for all other local minima $\theta_{1}$.*
*4.* 
*For k=0,1,2,3,4,5 the derivative $\partial_{\theta }^{k}L\left(\theta \right)$ exists, is continuous, and bounded on an open set around $\theta_{0}$.*
*5.* 
*For k=0,1,2,3,4,5, the variance $V[\partial_{\theta }^{k}\ell \left(\theta ,X_{n}\right)]\to 0$ as n→∞ on an open set around $\theta_{0}$.*


*Then, for sufficiently large n there exists a compact subset U⊂Θ containing $\theta_{0},\hat{\theta },$ such that:*
*1.* 
*For k=0,1,2,3 the derivative $\partial_{\theta }^{k}\ell^{*}\left(\theta ,x_{n}\right)$ exists, is continuous, and bounded on U, almost surely.*
*2.* 
*For k=0,1,2,3, $V[\partial_{\theta }^{k}\ell^{*}\left(\theta ,X_{n}\right)]\to 0$ as n→∞ on U, almost surely.*
*3.* 
*$\theta^{*}\in U$ as n→∞ almost surely.*
*4.* 
*$\sqrt{n}\left(\theta^{*}-\theta_{0}^{*}\right)\to N\left(0,{\left({\hat{J}}_{\theta_{0}^{*}}^{*}\right)}^{-1}{\hat{I}}_{\theta_{0}^{*}}^{*}{\left({\hat{J}}_{\theta_{0}^{*}}^{*}\right)}^{-1}\right)$ almost surely.*
*5.* 
*$-L\left(\theta^{*}\right)=-\ell^{*}\left(\theta^{*}\left(X_{n}\right),X_{n}\right)+O_{p}\left(n^{-3/2}\right)$ almost surely.*



**Proof.** See Appendix A of [1]. □

Theorem 1 would guarantee that $b\left(\theta^{*}\right)=O\left(\frac{m}{n^{3/2}}\right)$ if *I* and *J* could be known exactly.

Taking $v_{\left(\theta ,x\right)}=-\partial_{\theta }\ell \left(\theta ,x\right)$, and using the estimates $\hat{I}$ and $\hat{J}$ for their true values, this can be rewritten (approximately) as
(14)$$-\ell^{*}\left(\theta \right)=-\ell \left(\theta \right)+\frac{1}{n}\sum_{i}v_{\left(\theta ,x_{i}\right)}^{T}\left({\hat{J}}^{-1}\right)v_{\left(\theta ,x_{i}\right)}.$$

This substitution was also used by Takeuchi in [3].

Equation (Equation 14) (rather than Definition 7) will be the starting point for the analysis in the remainder of the paper. It is expected that since $\hat{I}\to I$ and $\hat{J}\to J$ this equation would converge to Definition 7 and $b\left(\theta^{*}\right)\to O\left(\frac{m}{n^{3/2}}\right)$; however, that will not be proven here since it is orthogonal to the analysis performed.

This paper is primarily concerned with the empirical consequences of approximating Equation (Equation 14) through various means. The consequence of using Equation (Equation 14) instead of Definition 7 is not directly observable or relevant to that analysis. As the analysis below makes clear, numerical instability would make any approach using the Hessian directly unviable, regardless of whether or not one could gain access to the actual true Hessian.

For background on the consequences of using an empirical approximation to the Hessian or Fisher Information in lieu of the actual unobservable value, see [14].

Experiments in [1] showed that the neighborhood of validity for this approach is typically large enough to contain $\theta^{*}$. Thus, if some care is taken with the optimization itself (techniques for this are also described in [1]), then this approach is quite widely applicable.

For realistic data set size *n*, reducing the generalization error *b* from $O(\frac{m}{n})$ to $O(\frac{m}{n^{3/2}})$ might effectively eliminate overfitting, without requiring any hyper parameters or cross validation. For an analysis of the scale of leading order bias terms, see [15], where it is seen in numerical simulations that first order corrections such as this can drastically reduce generalization errors.

Notice that for any models fit using ICE, it is sufficient to compare values of $-\ell^{*}\left(\theta \right)$ for model selection, as these are also valid approximations of TIC values. Both TIC and ICE approximate the log likelihood that would be computed using cross validation (if computed at $\hat{\theta }$, the MLE parameter estimate) and it can be seen that Equations (Equation 14) and (Equation 10) are identical.

As ICE is a superset of TIC, most of this paper will focus on ICE with the exception of sections comparing TIC to AIC.

The ICE approach, as with TIC, has a few main drawbacks.

Computation of $tr(\hat{I}{\hat{J}}^{-1})$ is expensive.Computation of $tr(\hat{I}{\hat{J}}^{-1})$ is numerically unstable.Since *J* must be positive definite, this is only valid in a neighborhood around the MLE.Computing derivatives of $\ell^{*}$ is expensive and potentially unstable.

Several proposals are made in [1] to overcome some of these issues. The purpose of this paper is to further analyze some of those proposals in the context of a large and more realistic problem, while also contributing additional improvements.

### 1.5. The Trace Term

The trace terms from Equations (Equation 10) and (Equation 14) are identical, and reproduced below.
(15)$$\frac{1}{n}\sum_{i}v_{\left(\theta ,x_{i}\right)}^{T}\left({\hat{J}}^{-1}\right)v_{\left(\theta ,x_{i}\right)}$$

Due to the inversion of $\hat{J}$, the computation of this term requires $O(m^{3})$ time (floating point operations) and $O(m^{2})$ space (bytes of memory) if *m* is the number of parameters in θ.

This equation is therefore well defined, but totally unsuitable for numerical computation, having two main problems:For even moderate parameter counts (e.g., 20+), the inverse condition number of $\hat{J}$ is typically less than machine precision. Therefore, the direct numerical computation of this quantity will be quickly dominated by numerical errors, even when a stabilized SVD-based pseudo-inversion is used.The computational cost of the inversion of $\hat{J}$ is $O(m^{3})$, and this must be performed on every iteration within the optimizer during model fitting. This quickly becomes intractable for parameter counts beyond a few hundred. This is less severe than the first issue, but still a major impediment to wide scale adoption in the highly parameterized models where bias is most an issue.

### 1.6. Efficient Approximations

In [1], several efficient approximations of Equation (Equation 14) have been proposed, and here we consider the following:(16)$$-\ell^{*}\left(\theta \right)=-\ell \left(\theta \right)+\frac{1}{n}\sum_{i}v_{\left(\theta ,x_{i}\right)}^{T}\left({\hat{D}}^{-1}\right)v_{\left(\theta ,x_{i}\right)}.$$
where here $\hat{D}$ is the diagonal of $\hat{J}$. In the analysis presented in [1], it was shown that using this approximation did not meaningfully impact accuracy, and even at times improved accuracy due to the numerical instability in the direct computation of ${\hat{J}}^{-1}$. Similar approximations are also used in other well-known numerical algorithms. For instance, the widely used ADAM [16] optimizer uses an even looser approximation by replacing *D* with the identity matrix.

### 1.7. Gradient Computation

To utilize TIC, the approximation in Equation (Equation 16) is sufficient; however, efficient computation of ICE also requires an approximate derivative. Since this derivative will be used only in optimizers, it need not be exact, but it is helpful if it generally has a high cosine similarity with the true derivative of Equation (Equation 16).

The gradient of the ICE objective function may be derived from Equation (Equation 14), and written as
(17)$$-\partial_{\theta }\ell^{*}\left(\theta \right)=v_{\left(\theta ,x\right)}+\partial_{\theta }\left[\frac{1}{n}\sum_{i}v_{\left(\theta ,x_{i}\right)}^{T}\left({\hat{J}}^{-1}\right)v_{\left(\theta ,x_{i}\right)}\right]$$
which simplifies to
(18)$$-\partial_{\theta }\ell^{*}\left(\theta \right)=v_{\left(\theta ,x\right)}+\frac{2}{n}\left[\sum_{i}J_{\left(\theta ,x_{i}\right)}{\hat{J}}^{-1}v_{\left(\theta ,x_{i}\right)}\right]+\frac{1}{n}\left[\sum_{i}v_{\left(\theta ,x_{i}\right)}^{T}{\hat{J}}^{-1}\left[\partial_{\theta }\hat{J}\right]{\hat{J}}^{-1}v_{\left(\theta ,x_{i}\right)}\right].$$

Notice that $J_{\left(\theta ,x_{i}\right)}{\hat{J}}^{-1}$ does not reduce to the identity, since it is a multiplication of the *J* matrix computed for a single observation against the inverse computed over all observations.

Direct computation of this quantity costs $O(m^{4})$ time, so an approximation is needed. Begin by applying the approximation from Equation (Equation 16) to obtain
(19)$$-\partial_{\theta }\ell^{*}\left(\theta \right)\approx v_{\left(\theta ,x\right)}+\frac{2}{n}\left[\sum_{i}J_{\left(\theta ,x_{i}\right)}{\hat{D}}^{-1}v_{\left(\theta ,x_{i}\right)}\right]+\frac{1}{n}\left[\sum_{i}v_{\left(\theta ,x_{i}\right)}^{T}{\hat{D}}^{-1}{\hat{D}}^{-1}\left[\partial_{\theta }\hat{D}\right]v_{\left(\theta ,x_{i}\right)}\right].$$

This is much improved, but still requires the computation of $\left[\partial_{\theta }\hat{D}\right]$, and $J_{\left(\theta ,x_{i}\right)}$, both of which cost $O(m^{2})$ in time and space. Further approximations are available to us here due to the fact that the optimizers that will use this gradient do not need its exact value. It is enough if the gradient is generally pointing “downhill” with respect to the objective function, and it is not necessary for it to be very accurate other than that. This translates to a requirement that the gradient approximation typically has a positive cosine-similarity with respect to the actual value.

If *n* is not too small, then the trace term is small compared to ℓ(θ), and $v_{\left(\theta ,x\right)}$ is small at $\theta^{*}$, but *D* is not. Similarly, $\left[\partial_{\theta }\hat{D}\right]$ need not be especially small or large in the neighborhood of $\theta^{*}$. Therefore, near $\theta^{*}$, the first correction term should be larger than the second, having only one factor of $v_{\left(\theta ,x_{i}\right)}$ instead of two.

Behavior far from $\theta^{*}$ would generally be less important than behavior near this optimum, since it is expected that ℓ(θ) would dominate these gradient terms in that case. Additionally, the numerical stabilization discussed in the next section (Equation (Equation 23)) will also tend to reduce the importance of any term containing *D* outside of the region where *D* is positive definite by forcing (in the limit) that J=D=I. In this limit, both of the correction terms would become a simple scalar multiple of $v_{\left(\theta ,x\right)}$, which would make them irrelevant to the optimization.

This reduces the equation to
(20)$$-\partial_{\theta }\ell^{*}\left(\theta \right)\approx v_{\left(\theta ,x\right)}+\frac{2}{n}\left[\sum_{i}J_{\left(\theta ,x_{i}\right)}{\hat{D}}^{-1}v_{\left(\theta ,x_{i}\right)}\right].$$

The computation here is still $O(m^{2})$, but one more approximation can be applied. Near $\hat{\theta }$, asymptotically, J=I under certain conditions. Therefore, making this substitution produces
(21)$$-\partial_{\theta }\ell^{*}\left(\theta \right)\approx v_{\left(\theta ,x\right)}+\frac{2}{n}\left[\sum_{i}v_{\left(\theta ,x_{i}\right)}\left[v_{\left(\theta ,x_{i}\right)}^{T}{\hat{D}}^{-1}v_{\left(\theta ,x_{i}\right)}\right]\right]$$
where now the inner quantity is the original ICE correction for this specific observation, and that is used as a weight for the unadjusted gradient. This quantity can then be stabilized using the techniques in Equation (Equation 23). Computationally, this is extremely efficient, requiring only O(m) time and space. This is one of the approaches that will be considered for gradient calculation.

The only remaining difficulty here is that this computation requires either two passes, or O(nm) space, because the matrix $\hat{D}$ must be computed first, and then applied element by element using Equation (Equation 21). Therefore, the gradients must either be computed twice (since they are used in the computation of $\hat{D}$), or their values stored. Alternatively, at a minor cost in accuracy, the $\hat{D}$ from the previous iteration could be used. This approach was used in the mortgage model examined here.

An alternative to the approximation in Equation (Equation 21) is to assume instead that $J_{\left(\theta ,x_{i}\right)}=D_{\left(\theta ,x_{i}\right)}$, here using the diagonal matrix $D_{\left(\theta ,x_{i}\right)}$ in place of the full $J_{\left(\theta ,x_{i}\right)}$. If that approximation is used, then
(22)$$-\partial_{\theta }\ell^{*}\left(\theta \right)\approx v_{\left(\theta ,x\right)}+\frac{2}{n}\left[\sum_{i}v_{\left(\theta ,x_{i}\right)}\left[D_{\left(\theta ,x_{i}\right)}{\hat{D}}^{-1}\right]\right].$$

This approximation may also be computed in O(m) time and space. However, this computation appears to be less stable than Equation (Equation 21), due to the likelihood of the non-positive definiteness of $D_{\left(\theta ,x_{i}\right)}$ for some observations $x_{i}$, and it will be seen in later sections that this is indeed the case.

The results section will analyze both Equations (Equation 21) and (Equation 22) numerically to determine which approach is more effective in this problem space.

### 1.8. Numerical Stabilization

Equations (Equation 16), (Equation 21) and (Equation 22) can all suffer from the potential singularity or ill conditioning of $\hat{D}$. This is a more severe problem for ICE than for TIC, since ICE must necessarily operate far from the MLE optimum $\hat{\theta }$ where *D* may be actually singular or not positive definite.

The analysis performed in [1] shows only that the trace term is the leading order generalization error term in a neighborhood *U* around $\hat{\theta }$, and need not be even positive outside of that neighborhood. Additional theories around the relationships between $\theta^{*}$ and $\hat{\theta }$ are not developed here, but should $\theta^{*}$ be close enough to $\hat{\theta }$ that it falls within *U*, and hence *J* is positive definite, then it is sufficient to ensure that the optimization over $-\ell^{*}\left(\theta \right)$ is able to reach *U*, and then within that neighborhood it can converge to $\theta^{*}$.

First of all, to improve numerical stability, we truncate to zero any gradient element with $∥{\left[v_{\left(\theta ,x_{i}\right)}\right]}_{k}∥\;<\sqrt{\varepsilon }*max_{k}(∥{\left[v_{\left(\theta ,x_{i}\right)}\right]}_{k}∥)$, where here ε is double precision machine epsilon, approximately $10^{-16}$. These terms are too small to change the outcome of a dot product within a machine error. A vector so truncated is indistinguishable via dot products from one that has not been; however, it is possible for such terms to add numerical instability due to rounding errors in the computation of ${\hat{D}}^{-1}$. Similarly, for each element ${\left[\hat{D}\right]}_{k}$ of $\hat{D}$, the value of ${\left[{\hat{D}}^{-1}\right]}_{k}$ is computed as
(23)$${\left[{\hat{D}}^{-1}\right]}_{k}\approx \frac{1}{w{\left[\hat{D}\right]}_{k}+\left(1-w\right){\left[v_{\left(\theta ,x_{i}\right)}\right]}_{k}^{2}}$$
where the weight *w* is computed as
(24)$$w_{k}=e^{-\frac{\sqrt{\varepsilon }max_{j}(∥{\left[\hat{D}\right]}_{j}∥)}{max(0,{\left[\hat{D}\right]}_{k})}}.$$

This weight is a continuous function of ${\left[\hat{D}\right]}_{k}$, and goes to zero as ${\left[\hat{D}\right]}_{k}$ becomes small enough that it is dominated by rounding errors. In addition, for negative values of ${\left[\hat{D}\right]}_{k}$, when multiplied by the square of the gradient in Equation (Equation 16), the term becomes 1.0, thus preventing instability from forming when the optimizer is not near the MLE solution and $\hat{D}$ is not positive definite.

Geometrically, this means that far outside of *U* the trace term is approximately the constant $\frac{m}{n}$ for sample size *n*, thus the optimizer will move towards *U* if the MLE optimization would have done so. As the optimizer draws closer to *U*, individual elements of the trace term start to take on values other than 1.0. Deep within *U*, the objective $-\ell^{*}\left(\theta \right)$ is essentially unchanged from the value that it would have in the absence of this correction, and thus the optimizer can freely converge to $\theta^{*}$. Proving that the adjustment from Equation (Equation 23) will always allow convergence to $\theta^{*}$ if $\theta^{*}\in U$ is beyond the scope of this paper. Qualitatively, this would be expected to usually work, and this behavior is analyzed numerically in later sections.

## 2. The Mortgage Modeling Problem and Dataset

The goal of this paper is to expand upon the techniques from [17], and apply them to larger datasets with more complicated models than the simplified ones analyzed there.

The real world data chosen for this analysis are a selection of Freddie Mac single family loan level mortgage data. These mortgages represent loans used to purchase or refinance single family homes within the United States. The data itself is available in [18], and can be found under the data_fit.dat and data_test.dat files. These loan records are a random subset originally pulled from the pre-2008 originations available from [19], and enriched to include Home Price Index (HPI) and Interest Rate (IR) data downloaded from FRED [20]. Pre-crisis (pre-2008) data were chosen since those loans have more data available for them (up to 12 years at the time of the data download) than post-crisis mortgages. The data themselves are described in Section A.1. These data were chosen because it is an open data set, contains a large amount of data, and is a problem of economic significance.

Each loan in the dataset is divided into a number of observations, one per month. In a given month, a loan may be in one of several states (i.e., status), and in the following month it may transition to a different status. Loan level mortgage models are classifiers used by major banks and other financial firms in order to evaluate the present value of individual mortgage loans. The behavior of the individual loans is aggregated to produce behavior projections for pools of loans, and the bonds collateralized by these pools. These bonds, combined with the pools themselves, are the primary instruments traded in the mortgage market, though individual underlying loans (known as whole loans) are also traded to a lesser degree. For a broad overview of mortgage finance, see [21].

The mortgage market as a whole encompasses approximately $3 trillion in valuation within the United States alone, making it one of the largest securities markets by value. This market is composed of approximately 60 million mortgages, and the market as a whole produces one new observation per month for each of these loans, so the dataset itself grows by roughly 60 million observations per month.

A complete mortgage modeling system would include one classifier for each loan status. Typically, this would be at least six classifiers, for statuses traditionally called C, 3, 6, 9, F, R, and there are two absorbing states, P and D, that do not need models. The goal of this system is to consider a loan in a given status (e.g., C) and predict its probability of transitioning to each of the other statuses. These transitions represent payments made or missed by the loan, so knowing the transitions produces a valuation for the loan. Typically, the next month’s status would be simulated using the calculated probabilities, and then the model would be run again using the simulated starting point for the next month. In this way, it would produce (one path of) the entire future trajectory of the loan, which implies all its cashflows.

For the calculations below, we consider only the most significant classifier, the classifier for loans that are current (status C) in the projected month. This is the set of loans that have made all required payments up until that point. A loan in status C can either make the required monthly payment this month (going to status C for next month), miss a payment (going to status 3) or refinance and liquidate (going to status P). These loans cannot miss multiple payments in a single month (because only one is due), so they cannot go to status 6 or 9, and they cannot be placed in foreclosure, so they cannot go to F, R, or D in the next month.

Having a highly accurate model for these transitions would allow a financial firm to accurately price mortgages, pools of mortgages, and bonds based on these pools, providing a competitive advantage within the mortgage market.

Two classifiers are analyzed in the results below.

The first classifier is an open sourced industrial model called ITEM [17]. This model was chosen because it is highly automated, greatly reducing questions of hyper parameters that might influence the results. It also makes heavy use of AIC, and so is a natural fit for discussions of information criteria. Additionally, the model is very parsimonious with parameters, making accurate models with only very few parameters. This production of low parameter count models makes the analysis of the direct computation of the trace term in Equation (Equation 14) viable. This direct computation will be used as a basis for comparison in the ITEM model section that follow, but will be omitted from the following neural network section due to unwieldy parameter counts.

The second classifier is a low depth Multilayer-Perceptron classifier [22]. This model is selected because it is very widely used. The parameter counts are too high to expect numerically stable computation of Equation (Equation 14), and therefore an approach due to LeCun [23] is used to approximate the diagonal entries of *J* in order to apply Equation (Equation 16).

## 3. Results Overview

The numerical results in the following sections are computed based on the mortgage dataset described in Section 2. For these sections, many different approximations and optimizations are used and tested, and it is necessary to show that each previous approximation is valid before moving on to later computations that will then rely on it. Therefore, the calculations here are divided into two sections.

First, the ITEM model is examined in Section 4. This model can accurately represent the problem with parameter counts small enough to enable direct computation of the trace term without approximations, and it is therefore used to compare those approximations to the direct computation. The ITEM section performs the following tests.

Establish direct trace computation encounters numerically singular $\hat{J}$ at $\hat{\theta }$: Section 4.2;Establish that numerical approximations correct the above issue: Section 4.2;Establish that the ICE approximations improve TIC even when used as an information criterion: Section 4.2;Establish which gradient computation is most accurate and measure its accuracy: Section 4.3;Establish that ICE (with optimizations and stabilization) outperforms MLE: Section 4.4;Establish that ICE is not unreasonably computationally expensive: Section 4.5.

With these analyses finished, it has been shown that the proposed ICE approach is numerically stable, efficient, and effective. Then, the approach is expanded to larger parameter counts (Section 4.6), to ensure that it does not fail as the parameter counts grow to several hundred parameters. This wraps up the section that uses the ITEM model.

Next, a Multilayer-Perceptron is examined in Section 5. This is a far more common model than ITEM, but since it is prone to very large parameter counts, it is necessary to use ITEM to establish the viability of the approach at smaller parameter counts first. Additionally, since this model uses an additional approximation attributed to LeCun [23], it is helpful to examine the accuracy of ICE without that approximation, in the ITEM context first. With that done, the Multilayer-Perceptron with LeCun’s approximation is shown to largely track what was found in ITEM in terms of the general utility of the ICE approach. The results section concludes with Section 5.3 establishing the reduction in generalization error produced by ICE for the Multilayer-Perceptron model.

## 4. Item Model

To show the viability of the ICE approach for real-world computations, a mortgage model was constructed from Freddie Mac single family loan level data. The model chosen here is ITEM, as described in [17]. This particular model was the basis for the US mortgage models at the Royal Bank of Canada starting from 2014 (Tyler Ward was the head of US mortgage modeling at RBC from 2014 to 2016). ITEM is an automated system for producing decompositions of datasets into analytic curves. It is chosen here due to its use in the industry for this problem, and because it uses AIC to automatically produce parsimonious models, and therefore direct computation using Equation (Equation 14) is viable as a basis for comparison. The raw output and code used for this section can be found in [18].

The ITEM procedure produces a sequence of models from a random seed, just as a Multilayer-Perceptron would produce different models for different random seeds due to the pseudo-random nature of the weight initialization. In this way, a large number of plausible models of varying parameter counts can be generated. Statistics can be collected describing what a typical operator would be likely to obtain from each of the estimation methods considered.

To implement this in a real-world situation, two main difficulties must be overcome.

The ICE objective must be computed efficiently.Approximate gradients must be computed efficiently.

Approximation formulas were developed above in Equations (Equation 16), (Equation 21), and (Equation 22).

### 4.1. Model Construction

In the following sections, the ITEM model generation procedure was allowed to run from various starting seeds. For the parameter estimation approaches used, see Table 1.

For each of these models, 20 series of ITEM models (100 series total) were generated. Each series produced several models as larger models were iteratively constructed from smaller ones, see [17] for a description of how this is performed. The analysis then considered all intermediate and final models from each series, for a total of 2123 models with parameter counts ranging from 2 to 83.

#### L2 Regularization

$L_{2}$ regularization (i.e., ridge-regression) was attempted with various values of λ, but all values of λ≠0 were found to be harmful or statistically indistinguishable from zero. Consequently, these models were not qualitatively different from the MLE series described above. This was an expected outcome, and these approaches were not considered further in this paper.

For an example of why λ≠0 was not helpful, see the results in Section 4.2 of [1], and note that the ITEM models generated here also use standard deviation-like parameters, and so suffer from the same problems with $L_{2}$ regularization. Additionally, the scale of the regressors to this model covers more than 9 orders of magnitude (e.g., unpaid balance is of order $10^{7}$ and incentive is of order $10^{-2}$), greatly hampering $L_{2}$ and LASSO-like approaches unless data normalization is applied as a preprocessing step.

### 4.2. Numerical Stability of TIC and ICE

To evaluate the numerical stability of ICE and TIC, we found the MLE parameter estimate (i.e., $\hat{\theta }$) for each of the 2123 models, and then computed the inverse condition number of $\hat{J}$ and $\hat{I}$ at each of these optimum points. For this analysis, the matrix $\hat{J}$ is numerically singular whenever its inverse condition number is less than $\sqrt{\varepsilon }$, with ε being machine precision, approximately $10^{-16}$ for double precision floating point.

Within the literature, proper analysis of numerical stability of this term is severely lacking. Kitagawa and Konishi performed some simulations in [15], but only for m=2. Therefore, they did not encounter difficulty inverting *J*, which (in the simulations below) begins to rapidly escalate for m>10. Section 2.3 of Burnham and Anderson [9] gives the numerical instability of TIC when m≥20 as a primary reason for avoiding it, and for it not seeing widespread use.

Additional analysis of the numerical stability of TIC and related methods can be found in [8], where Section 3 provides a good overview of recent research in the area, and some numerical results for approximations of *J*. The computation of an accurate inverse Hessian ($J^{-1}$) is a persistent source of difficulty in machine learning.

In this circumstance, these approaches are being used as information criteria since they are evaluated at $\hat{\theta }$. For clarity, the approaches are listed here as information criteria in Table 2.

The approaches TIC2 and TIC3 are applying to TIC the approximations developed for ICE.

When $\hat{J}$ is numerically singular, we expect that ${\hat{J}}^{-1}$ has unstable behavior and TIC (the only approach directly using ${\hat{J}}^{-1}$) becomes unstable. The other approaches based on Equation (Equation 16) are not expected to be affected by this, since a diagonal matrix may be safely pseudo-inverted regardless of its condition number.

The numerical instability displayed in Table 3 indicates that TIC when directly computed will not be numerically stable, in addition to its $O(m^{3})$ computational cost.

To verify the numerical stability of Equations (Equation 14) and (Equation 16), we compute the size of the adjustment term ($tr(\hat{I}{\hat{J}}^{-1}))$, and compare it to the parameter count (*m*), which is its asymptotic limit if the model is correctly specified, see [4]. The results are labeled according to the approach as described in Table 2.

In Table 4, it can be seen that the TIC3 approach is more clustered near 1.0 than either of the other approaches. The direct TIC approach shows signs of instability for higher parameter counts where the average adjustment turns negative at the MLE estimate $\hat{\theta }$ where $\hat{J}$ should be positive definite. This is due to the instability described in Table 3. The approach TIC2 has large swings in value through the parameter space, and between models within each of these groups, indicating that it too is suffering from instability that the TIC approach is able to correct using the methodology described in Equation (Equation 23).

In Table 5, it is shown that TIC and TIC2 have a significant proportion of models with a negative objective function adjustment, even though $\hat{J}$ should be positive definite at $\hat{\theta }$. This indicates substantial numerical instability with these approaches. The approach TIC3 does not have any negative adjustments, in fact the lowest scaled adjustment (in the sense of Table 4) among any of the 2123 models considered here is 0.53.

The conclusion that can be drawn here is that TIC is inappropriate for direct use for many models with even moderate parameter counts. However, applying the approximations and numerical stabilization described in [1] corrects many of these issues and might allow for a wider application of this adjusted approach to TIC. The additional numerical stabilization from Equation (Equation 23) is necessary in order to achieve reasonable results.

In Table 6, the sum of absolute errors is shown for each group of models between the test set value of $\ell (\hat{\theta })$ and the information criterion computed for AIC, TIC, and TIC3. Curiously, TIC does worst for relatively lower parameter counts. Its results have higher variability for higher parameter counts, but the mean is not appreciably different from AIC on average. These results are, however, highly unstable for TIC and thus not appropriate for model selection.

Though the approach ICE_A was used to produce approximately 1/5 of the 2123 models considered, it is clear from the analysis above that it is substantially worse than the ICE approach (TIC3 in the information criterion context), and thus it need not be considered further.

### 4.3. Gradient Accuracy

The finite-difference gradients of Equation (Equation 16) with stabilization via Equation (Equation 23) for the 2123 target models were computed. These were then compared to the approximate gradients computed using Equations (Equation 21) and (Equation 22). The average cosine similarity was computed, and also the fraction of gradients with positive cosine similarity, summarizing the results in Table 7.

This analysis finds good agreement between the finite difference derivatives and the approximation through Equation (Equation 21), and somewhat lesser agreement using Equation (Equation 22). This indicates that approach ICE from Table 1 may outperform approach ICE_B.

### 4.4. Prediction Accuracy

For the 2123 target models, we refit the parameters of each using each of the target methodologies, and then compute the out of sample cross entropy. The results are summarized in Table 8. Here, the approach ICE_A has been eliminated, but ICE_RAW has been retained to represent the numerically unstable approaches and their impact on accuracy.

As can be seen in Table 8, the ICE and ICE_B approaches perform similarly. This is expected, since they share the same objective function. Both of these approaches are a small improvement on MLE overall for every parameter group listed here. The approach ICE_RAW performs very badly due to its numerical instability for parameter counts larger than 20, and the performance of ICE_A (not shown) is similar. The individual entropy differences are small, but the outperformance of ICE is highly significant.

As can be seen in Table 9, ICE outperforms MLE at the 4 sigma level throughout most of the parameter range. The ICE_RAW approach has a comparatively large standard deviation due to numerical instability, but it underperforms at the 2 sigma level for most parameter counts.

### 4.5. Computational Performance

For each of the target models, we recorded how long (in ms) a computer required to perform a parameter fit. The results are summarized in Table 10. The MLE estimates are omitted, because the optimization starts at $\hat{\theta }$ so this optimization is vacuous.

As can be seen in Table 10, the cost of the ICE_RAW methodology appears to be super linear in the parameter count. The other ICE methodologies are more linear, though with considerable noise. Much of this variation is due to the optimizers requiring more or fewer objective function evaluations to find an optimum. Therefore, some approaches could be more expensive due to poor derivatives causing more evaluations to be needed in order to find the optimum. To correct for this, we measure the time required for a computer to evaluate the objective function for each of these approaches. The time required is divided by the parameter count for each model, and then the results are grouped by parameter count and averaged. The results are summarized in Table 11.

As can be seen from Table 11, since ICE and ICE_B share an objective function, they have equivalent costs. In both cases, the cost is highly linear with the parameter count, as expected. For these parameter sizes, the MLE objective cost is dominated by exponentials and logarithms in the multinomial logistic and entropy calculations, respectively, and the MLE objective does grow sub-linearly through this parameter range. The ICE_RAW objective function cost is dominated by the $O(m^{3})$ inversion of $\hat{J}$, and in this parameter range its cost grows super linearly. ICE_RAW and ICE begin at nearly the same cost, but for models with at least 30 parameters ICE_RAW is already more than twice as expensive as ICE. We expect that for larger parameter counts, the cost of ICE_RAW would be prohibitive.

The computational cost of each gradient computation was measured, and the results are summarized in Table 12. Again, each value was divided by the parameter count, as we expect all of these approaches to be O(m).

In Table 12, the gradient computations for ICE, ICE_B, and ICE_RAW are almost exactly 4 times the cost of MLE throughout the entire parameter range. The approach ICE_RAW uses the same gradient computation as ICE, and ICE_B should have nearly identical computational complexity. We see here that this is indeed the case. For most model optimization approaches, gradient computation is the limiting factor. For such approaches, the ICE methodologies incur a small constant multiple in fitting costs.

Additional performance improvements are beyond the scope of the present work, but the current implementation is not efficiently sharing computations needed to produce $v_{\left(\theta ,x_{i}\right)}$ and those needed to compute $\hat{D}$. In a more tuned implementation, the cost of ICE could be reduced by a small constant factor.

### 4.6. Large Model Accuracy

The most accurate (out of sample) model was selected from among the target models. This model had 32 parameters and an out of sample entropy of 0.19677. That model was then repeatedly expanded in order to generate a sequence of overfit models.

For each regressor, for each quantization, from 3 to 10 buckets was produced. Then, for each such bucket except the first and last from each quantization, a Gaussian and Logistic curve was added centered on that bucket. For each such curve, the width of the curve (std. dev. of the Gaussian, and inverse slope for the Logistic) was chosen to match the width of the target bucket. This procedure produced a sequence of 83 models having between 32 and 764 parameters. Each such model was fit using MLE, and then re-fit using each of MLE, ICE and ICE_B to examine the quality of bias reduction in the case of overfit models. Many of these models settled on local minima and thus exhibited considerably worse performance than the baseline model. The results were grouped by the parameter count and then summarized in Table 13.

The T-statistics of the differences between the ICE and ICE_B and MLE models are summarized in Table 14.

The ICE_B approach may have statistically significant degradation in performance for large models (more than 300 parameters), but the ICE methodology itself does not appear to suffer from this. Some of its most significant results are actually produced for comparatively large parameter counts. Similarly, in Table 13, it can be seen that many of the largest improvements in absolute magnitude for the ICE model are also produced at relatively high parameter counts. None of these values are significant at the 5 sigma level, and only three of them are significant at the 3 sigma level (all in favor of ICE and ICE_B).

## 5. Neural Network Implementation

For implementation within neural networks, it is necessary to be able to compute $\hat{D}$ using back-propagation. The techniques for performing this computation are described by LeCun in Section 3.2 of [23]. Another description of this approach is given in Section 4.1 of [24].

Consider the neural network with *L* layers, and cost function *C*. Assume also that no connections skip layers. Typically, for a classifier, *C* would be a cross entropy loss, with *y* being the known labels of the training data.
(25)$$C(y,f_{L}\left(W_{L}f_{L-1}\left(W_{L-1}\ldots f_{1}\left(W_{1}x\right)\right)\right)))).$$

Here, $W_{l}$ is the matrix of edge weights for layer *l*, and it is assumed that the activation function at each layer is uniform, hence univariate function $f_{l}$ when applied to a vector argument produces a vector output. We may define the activation of layer *l* as
(26)$$a_{0}=x,$$
(27)$$a_{l}=f_{l}\left(W_{l}f_{l-1}\left(W_{l-1}\ldots f_{1}\left(W_{1}x\right)\right)\right))\;=\;f_{l}\left(W_{l}a_{l-1}\right).$$

Then, we can rewrite the neural network, ignoring the parameter *y*, as
(28)$$C(a_{L}).$$

Note that the weights contain an implicit bias term, so more explicitly, the activation of the *i*’th node in layer *l* would be
(29)$${\left(a_{l}\right)}_{i}=f_{l}\left({\left(W_{l}\right)}_{\left(i,0\right)}+\sum_{k}{\left(a_{l-1}\right)}_{k}{\left(W_{l}\right)}_{\left(i,k\right)}\right).$$

Then, the second derivative of the objective function *C* of the network may be constructed by inverting this sum (so it runs over *i* that is connected to by *k*).
(30)$$\frac{\partial^{2}C}{\partial {\left(a_{l-1}\right)}_{k}^{2}}=\sum_{i}\left[\frac{\partial^{2}C}{\partial {\left(a_{l}\right)}_{i}^{2}}{\left({\left(f_{l}^{\prime }\left(W_{l}a_{l-1}\right)\right)}_{i}{\left(W_{l}\right)}_{\left(i,k\right)}\right)}^{2}+\frac{\partial C}{\partial {\left(a_{l}\right)}_{i}}{\left(f_{l}^{\prime \prime }\left(W_{l}a_{l-1}\right)\right)}_{i}{\left(W_{l}\right)}_{\left(i,k\right)}^{2}\right]$$
where here the derivatives $f_{l}^{\prime }$ and $f_{l}^{\prime \prime }$ are taken with respect to the function’s sole argument. Note that this equation is only accurate for the case where the off diagonal elements of $\frac{\partial^{2}C}{\partial {\left(a_{l-1}\right)}_{k}^{2}}$ are actually zero. If any are nonzero (as would be the case in practice), then this equation is only an approximation. Renaming this quantity
(31)$${\left(u_{l}^{\prime }\right)}_{i}=f_{l}^{\prime }\left(W_{l}a_{l-1}\right)$$
and
(32)$${\left(u_{l}^{\prime \prime }\right)}_{i}=f_{l}^{\prime \prime }\left(W_{l}a_{l-1}\right)$$

Equation (Equation 30) may be rewritten as
(33)$$\frac{\partial^{2}C}{\partial {\left(a_{l-1}\right)}_{k}^{2}}=\sum_{i}\left[\frac{\partial^{2}C}{\partial {\left(a_{l}\right)}_{i}^{2}}{\left(u_{l}^{\prime }\right)}_{i}^{2}+\frac{\partial C}{\partial {\left(a_{l}\right)}_{i}}{\left(u_{l}^{\prime \prime }\right)}_{i}\right]{\left(W_{l}\right)}_{\left(i,k\right)}^{2}.$$

The derivatives with respect to the weights are then
(34)$$\frac{\partial^{2}C}{\partial {\left(W_{l}\right)}_{\left(i,k\right)}^{2}}=\left[\frac{\partial^{2}C}{\partial {\left(a_{l}\right)}_{i}^{2}}{\left(u_{l}^{\prime }\right)}_{i}^{2}+\frac{\partial C}{\partial {\left(a_{l}\right)}_{i}}{\left(u_{l}^{\prime \prime }\right)}_{i}\right]{\left(a_{l-1}\right)}_{k}^{2}.$$

These formulas are then suitable for a back-propagation implementation.

### 5.1. Back-Propagation Implementation

For modern neural nets, derivatives must be computed using back-propagation for efficiency reasons. This section describes the back-propagation techniques used to compute first and second derivatives.

#### 5.1.1. Back-Propagation Gradient Implementation

Considering the network as previously defined, we may define the auxiliary value
(35)$$\delta_{L}=\left(u_{L}^{\prime }\right)\cdot \nabla_{a_{L}}C$$
and then recursively define it for all other layers.
(36)$$\delta_{l-1}=\left(u_{l-1}^{\prime }\right){\left(W_{l}\right)}^{T}\delta_{l}.$$

We then may compute the gradient of *C* using these values.
(37)$$\nabla_{W_{l}}C=\left(\delta_{l}\right){\left(a_{l-1}\right)}^{T}.$$

For future reference, note that
(38)$$\frac{\partial C}{\partial (a_{l-1})}=\nabla_{a_{l-1}}C=\left(\delta_{l}\right){\left(W_{l}\right)}^{T}$$
and that $\frac{\partial C}{\partial (a_{L})}$ is directly computable from the definition of *C*.

#### 5.1.2. Back-Propagation Hessian Diagonal Implementation

Analogously to the definition of δ, we define an auxiliary value γ using Equation (Equation 30). Because this will be computing only the diagonal of the Hessian, it is necessary to write it in summation form.
(39)$$\begin{array}{ccc} {\left(\gamma_{l-1}\right)}_{k} & = & \frac{\partial^{2}C}{\partial {\left(a_{l-1}\right)}_{k}^{2}} \end{array}$$
(40)$$\begin{array}{ccc} & = & \sum_{i}\left[\frac{\partial^{2}C}{\partial {\left(a_{l}\right)}_{i}^{2}}{\left(u_{l}^{\prime }\right)}_{i}^{2}+\frac{\partial C}{\partial {\left(a_{l}\right)}_{i}}{\left(u_{l}^{\prime \prime }\right)}_{i}\right]{\left(W_{l}\right)}_{\left(i,k\right)}^{2} \end{array}$$
(41)$$\begin{array}{ccc} & = & \sum_{i}\left[{\left(\gamma_{l}\right)}_{i}{\left(u_{l}^{\prime }\right)}_{i}^{2}+\frac{\partial C}{\partial {\left(a_{l}\right)}_{i}}{\left(u_{l}^{\prime \prime }\right)}_{i}\right]{\left(W_{l}\right)}_{\left(i,k\right)}^{2} \end{array}$$
and that $\left(\gamma_{L}\right)=\frac{\partial^{2}C}{\partial {\left(a_{L}\right)}^{2}}$ is computable directly from the definition of *C*. Additionally, $\frac{\partial C}{\partial {\left(a_{l}\right)}_{i}}$ may be computed using Equation (Equation 38).

Then, the diagonal of the Hessian itself is computed using Equation (Equation 34).
(42)$$\frac{\partial^{2}C}{\partial {\left(W_{l}\right)}_{\left(i,k\right)}^{2}}=\left[{\left(\gamma_{l}\right)}_{i}{\left(u_{l}^{\prime }\right)}_{i}^{2}+\frac{\partial C}{\partial {\left(a_{l}\right)}_{i}}{\left(u_{l}^{\prime \prime }\right)}_{i}\right]{\left(a_{l-1}\right)}_{k}^{2}.$$

The combination of Equations (Equation 41), (Equation 42), and (Equation 37) are sufficient to compute the first and (non-mixed) second derivatives of the neural network in a single back-propagation pass.

Recall that Equation (Equation 30) is only an approximation. If more accuracy is needed (at the expense of more computation), then the matrix ${\left(\Gamma_{l}\right)}_{\left(i,k\right)}$ (instead of the vector ${\left(\gamma_{l}\right)}_{i}$) may be computed and back-propagated using a similar formula. In which case Equation (Equation 42) relies instead on ${\left(\Gamma_{l}\right)}_{\left(i,i\right)}$, but is otherwise unchanged. That analysis is beyond the scope of the current work.

### 5.2. Derivatives of Cross-Entropy Multinomial Logistic Loss

Suppose the loss function *C* is cross-entropy loss using a multinomial logistic (i.e., softmax) classifier. Defining the vector valued multinomial logistic function as
(43)$${\left(L\left(a_{L}\right)\right)}_{i}=\frac{exp({\left(a_{L}\right)}_{i})}{\sum_{k}exp\left({\left(a_{L}\right)}_{k}\right)}.$$

Then, the cross entropy loss of a single observation is
(44)$$C{\left(y,\left(a_{L}\right)\right)}_{i}=-y_{i}\ln\left[\frac{exp({\left(a_{L}\right)}_{i})}{\sum_{k}exp\left({\left(a_{L}\right)}_{k}\right)}\right]=-y_{i}\ln\left[{\left(L\left(a_{L}\right)\right)}_{i}\right]$$
where *y* is a one-hot encoding of the classes for the given observation. The derivatives of this loss function with respect to $a_{L}$ are
(45)$$\frac{\partial C(y,a_{L})}{\partial {\left(a_{L}\right)}_{i}}=\left(L{\left(a_{L}\right)}_{i}-y_{i}\right)$$
and
(46)$${\left(\frac{\partial^{2}C}{\partial {\left(a_{L}\right)}_{i}^{2}}\right)}_{i}=\frac{\partial }{\partial {\left(a_{L}\right)}_{i}}\left[L{\left(a_{L}\right)}_{i}-y_{i}\right]=\left[1-L{\left(a_{L}\right)}_{i}\right]L{\left(a_{L}\right)}_{i}.$$

Note that traditionally, a Multilayer-Perceptron will use the identity activation function for the last layer, in which case $f_{L}\left(x\right)=x$.

### 5.3. Prediction Accuracy

The ICE estimator was implemented in the Apache Spark MultilayerPerceptronClassifier, and compared against a stock MultilayerPerceptronClassifier using Spark version 2.4.5. This implementation was chosen due to the dominant marketshare of Spark and the ease of implementation and testing within that codebase. Because the Spark MLP model does not provide for regularization or drop-out, this approach could not be compared against those approaches within this codebase.

The computation was performed on the same data set used for the ITEM calculations above, described in Section A.1.

For this section, accuracy was tested on four layer configurations. Each model has 11 input regressors and three classification states. The models tested are described in Table 15. The data were standardized before applying the neural network, as is common practice. The objective function used Equation (Equation 16) with Equation (Equation 23), and used LeCun’s approximation (i.e., Equation (Equation 42)) to compute *D*, the diagonal of *J*. The gradients were computed using Equation (Equation 21). With the exception of the addition of LeCun’s approximation, this is the same setup as was used for the models labeled ICE above.

Each configuration was fit 10 times on randomly drawn fitting sets of various sizes using both MLE and ICE. The cross-entropy on the testing set was averaged for each series of tests. All optimizations are performed using l-bfgs, which generally produced better fits in all the tests. The results are presented in Figure 1, Figure 2, Figure 3 and Figure 4.

The models given here are relatively small compared to the huge models often found in machine learning (e.g., see [8]), but they are still much larger than the very small models with a handful of parameters often analyzed in statistical research. Whether or not these approaches generalize to much larger models with thousands or millions of parameters is beyond the scope of this present work, but there is no impediment found here that would prevent wider application to larger models.

In all four configurations, ICE effectively eliminates overfitting for all but the smallest sample sizes, whereas MLE suffers severe overfitting for smaller sample sizes. In all four tests, MLE performs slightly better with very large sample sizes, but the difference is not large. For the [11,3] configuration shown in Figure 1, ICE shows some overfitting, but much less than MLE. For the other configurations, no material amount of overfitting is present. This is likely due to the specifics of the l-bfgs fitting algorithm, which can generally search the parameter space much more efficiently for a more nearly linear model such as [11,3] than it can for more complicated configurations. The bias reduction from ICE is asymptotic, so it is not surprising that the approach is weaker with very small sample sizes. For larger models with correspondingly larger sample sizes, ICE is more consistently helpful. Regardless, ICE still greatly outperforms MLE for small sample sizes in even this smaller model.

For all four configurations, ICE fitting time (not shown) was not materially different from the time required to fit with MLE. The computation of the ICE loss and gradient as described here theoretically requires a small constant factor more computation than MLE loss and gradients. For these tests, both costs are swamped by other factors and overheads.

For the neural network model described here, a more full analysis of the TIC term numerical instability cannot be performed due to the reliance on LeCun’s approximation, which provides only the diagonal matrix *D*, not the full Hessian *J*. Additionally, the inversion of *J* would be overly costly at these parameter counts, therefore those analyses are available only in Section 4.

## 6. Conclusions

It was shown in this paper that for a real world mortgage model in use within the industry, the incorporation of ICE can substantially improve the prediction accuracy at the cost of a small constant multiple increase in fitting time. Additionally, it was shown that the approach described by [1] can be successfully implemented in a Multilayer-Perceptron, and should be applicable to any back-propagating neural network using the techniques described here.

The numerical stability of the TIC term from [3] was explored on a real world problem using industrial models. Numerical approximations and stabilization techniques were demonstrated that greatly reduce the effect of numerical instability for this term, which might otherwise prevent the application of TIC for problems with a significant number of parameters.

The diagonal approximation of LeCun was shown to produce good results in networks of reasonable depth, and may serve as the basis for the application of ICE and stabilized TIC techniques to a broad class of neural networks of moderate depth.

## Figures and Tables

**Figure 1 entropy-25-00512-f001:**
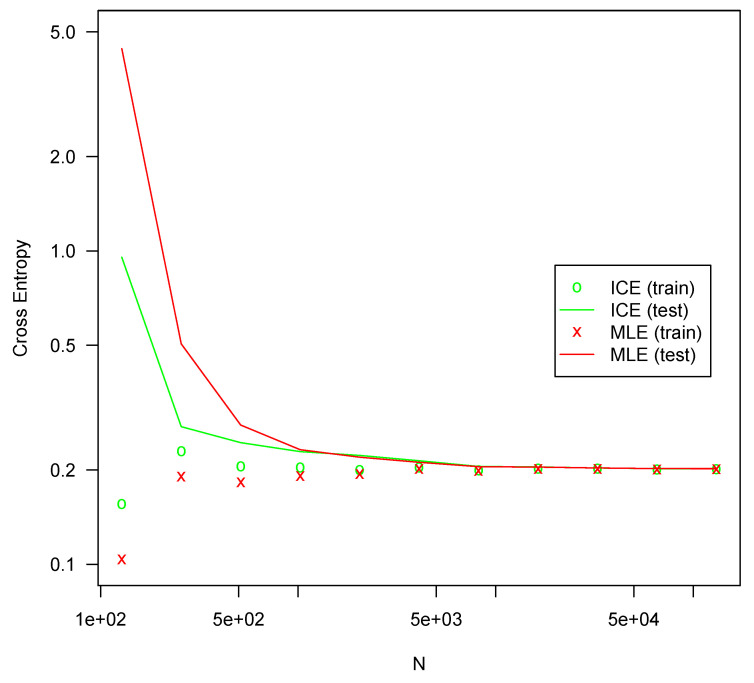
Cross entropy loss for configuration [11,3] (36 parameters).

**Figure 2 entropy-25-00512-f002:**
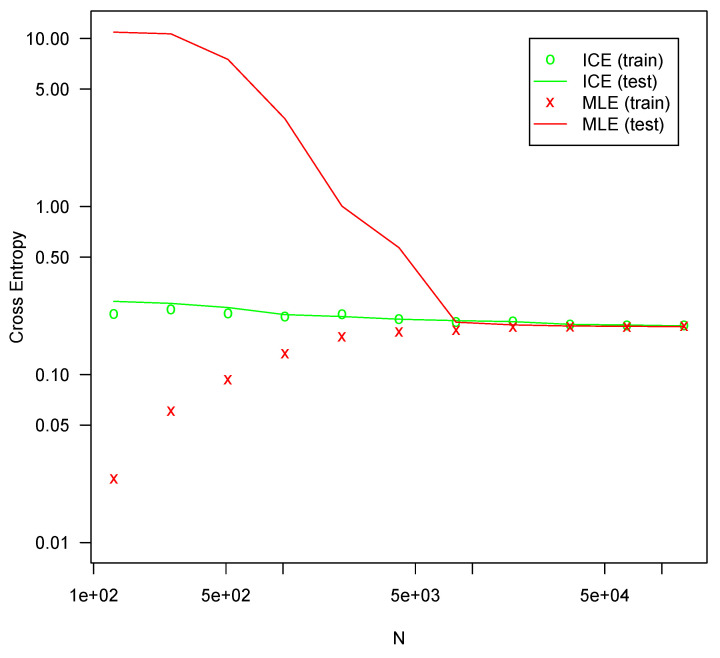
Cross entropy loss for configuration [11,5,3] (78 parameters).

**Figure 3 entropy-25-00512-f003:**
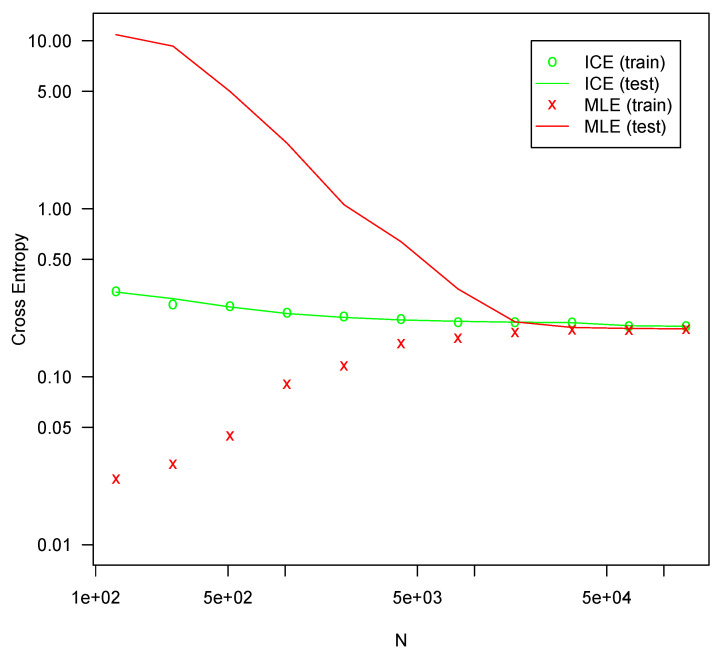
Cross entropy loss for configuration [11,8,5,3] (159 parameters).

**Figure 4 entropy-25-00512-f004:**
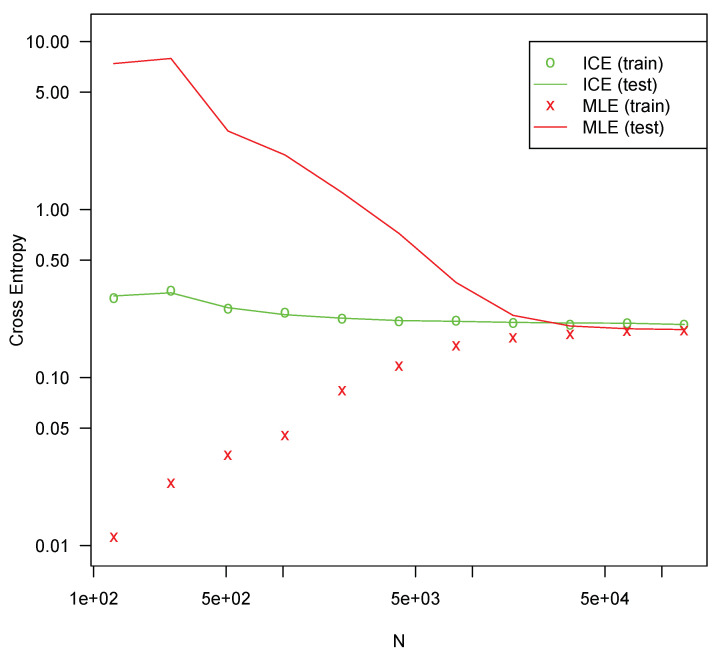
Cross entropy loss for configuration [11,11,8,5,3] (291 parameters).

**Table 1 entropy-25-00512-t001:** Estimation approaches compared.

Name	Objective Function	Gradient Approximation
MLE	ℓ(θ)	Exact Analytic Gradients
ICE_RAW	Equation (Equation 14)	Equation (Equation 21)
ICE	Equation (Equation 16) with Equation (Equation 23)	Equation (Equation 21)
ICE_A	Equation (Equation 16)	Equation (Equation 21)
ICE_B	Equation (Equation 16) with Equation (Equation 23)	Equation (Equation 22)

**Table 2 entropy-25-00512-t002:** Information criteria compared.

Name	Calculation
AIC	Equation (Equation 11)
TIC	Equation (Equation 14)
TIC2	Equation (Equation 16)
TIC3	Equation (Equation 16) with Equation (Equation 23)

**Table 3 entropy-25-00512-t003:** The fraction of $\hat{J}$ that are numerically singular, and actually singular by parameter count.

Parameter Range	Model Count	Numerically Singular $\hat{J}$	Singular $\hat{J}$
0–9	529	0.06	0.00
10–19	356	0.71	0.02
20–29	459	0.98	0.14
30+	779	1.0	0.16

**Table 4 entropy-25-00512-t004:** The objective function adjustment divided by parameter count (i.e., $\frac{1}{m}tr\left(\hat{I}{\hat{J}}^{-1}\right)$) for several objective functions, grouped by parameter count. Individual models have their computations bounded between −100 and 100 for TIC, as otherwise those results are driven by a few severe outliers.

Parameter Range	Model Count	TIC	TIC2	TIC3
0–9	529	−1.13	0.84	1.09
10–19	356	−5.86	−0.01	0.97
20–29	459	−4.63	0.68	0.85
30+	779	−4.58	0.96	0.79

**Table 5 entropy-25-00512-t005:** The fraction of models having negative objective function adjustment at $\hat{\theta }$, grouped by parameter count.

Parameter Range	Model Count	TIC	TIC2	TIC3
0–9	529	0.09	0.01	0.00
10–19	356	0.25	0.03	0.00
20–29	459	0.19	0.03	0.00
30+	779	0.21	0.03	0.00

**Table 6 entropy-25-00512-t006:** The mean of absolute errors between test set cross entropy and information criterion. Maximum value in parentheses.

Parameter Range	Model Count	AIC	TIC	TIC3
0–9	529	0.0026 (0.0036)	0.0563 (28.4430)	0.0026 (0.0036)
10–19	356	0.0030 (0.0037)	0.0112 (2.8471)	0.0030 (0.0037)
20–29	459	0.0030 (0.0036)	0.0029 (0.0084)	0.0030 (0.0041)
30+	779	0.0031 (0.0038)	0.0030 (0.0743)	0.0030 (0.0037)

**Table 7 entropy-25-00512-t007:** Comparison gradients using analytical approximations with their finite difference values.

Approximation Used	Cosine Similarity	Positive Fraction
ICE (Equation (Equation 21))	0.73	0.91
ICE_B (Equation (Equation 22))	0.54	0.75

**Table 8 entropy-25-00512-t008:** Out of sample cross entropy of fitting methodologies.

Parameter Range	Model Count	MLE	ICE	ICE_B	ICE_RAW
0–9	529	0.20813	0.20812	0.20812	0.20815
10–19	356	0.20056	0.20055	0.20055	0.20065
20–29	459	0.19851	0.19849	0.19851	0.19875
30+	779	0.19725	0.19722	0.19725	0.19806

**Table 9 entropy-25-00512-t009:** T-statistics of differences between given methodology and MLE out of sample cross entropy (negative indicates the approach is performing better).

Parameter Range	Model Count	ICE	ICE_B	ICE_RAW
0–9	529	−4.472	−3.79	3.83
10–19	356	−1.89	−0.91	5.96
20–29	459	−4.56	1.06	2.68
30+	779	−8.18	−0.37	1.43

**Table 10 entropy-25-00512-t010:** Average computational cost (in ms) of parameter optimization starting from $\hat{\theta }$.

Parameter Range	Model Count	ICE	ICE_B	ICE_RAW
0–9	529	573	592	592
10–19	356	2548	2487	3563
20–29	459	2611	2705	5363
30+	779	3166	3349	9263

**Table 11 entropy-25-00512-t011:** Cost (in ms) per evaluation of the objective function divided by the parameter count.

Parameter Range	Model Count	MLE	ICE	ICE_B	ICE_RAW
0–9	529	6.88	23.49	23.63	32.98
10–19	356	2.85	23.27	23.19	46.94
20–29	459	2.29	22.00	21.94	55.87
30+	779	1.97	21.14	21.43	76.25

**Table 12 entropy-25-00512-t012:** Cost (in ms) per evaluation of the objective function gradient divided by the parameter count.

Parameter Range	Model Count	MLE	ICE	ICE_B	ICE_RAW
0–9	529	23.67	97.51	97.55	97.82
10–19	356	23.17	95.32	95.51	95.32
20–29	459	21.97	90.47	90.47	90.57
30+	779	21.45	88.48	88.45	88.36

**Table 13 entropy-25-00512-t013:** Out of sample performance of fitting methodologies for overfit models.

Parameter Range	Model Count	MLE	ICE	ICE_B
32–199	37	0.19766	0.19753	0.19755
200–299	16	0.20394	0.20298	0.20224
300–399	13	0.21794	0.21423	0.21604
400–499	7	0.23772	0.23360	0.23302
500–599	5	0.25508	0.25114	0.25316
600–699	3	0.29203	0.28052	0.30099
700+	2	0.26392	0.26833	0.25448

**Table 14 entropy-25-00512-t014:** T-statistics of differences between the given methodology and MLE out of sample cross entropy (negative indicates the approach is performing better) for large models.

Parameter Range	Model Count	ICE	ICE_B
32–199	37	−1.47	−1.51
200–299	16	−1.45	−3.12
300–399	13	−3.62	−1.86
400–499	7	−1.67	−1.28
500–599	5	−0.54	−0.72
600–699	3	−3.17	2.95
700+	2	1.19	−0.49

**Table 15 entropy-25-00512-t015:** The model configurations.

Layer Configuration	Parameter Count	Description
[11,3]	36	The simplest model, with no hidden layers.
[11,5,3]	78	A model with a single 5 wide hidden layer.
[11,8,5,3]	159	A model with two hidden layers.
[11,11,8,5,3]	291	A model with three hidden layers.

## Data Availability

Data available at https://doi.org/10.6084/m9.figshare.20751181.v1.

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
