# Peer review of "Improving the Performance and Stability of TIC and ICE"

_entropy, 2023, doi:10.3390/e25030512_

Round 1
Reviewer 1 Report (Previous Reviewer 3)
The author has satisfactorily addressed my comments. I now deem the paper suitable for publication.
Author Response
Thank you for your review.
Reviewer 2 Report (Previous Reviewer 1)
The manuscript still has many room for improvement. In particular, the description of ICE method is hard to understand and not convincing enough, at least for statisticians. One reason is that it is not written in Definition-Theorem style. Such a mathematical clarification is necessary to clarify the technical detail of ICE method. Also, some professional editing of English is desirable.
Author Response
Thank you for the review.
In response to your concerns about the ICE results needing further clarification, I did the following.
1) Added more clarifying material.
2) Changed some of the equations to definitions in cases where they really are properly definitions.
3) Brought in a Theorem (really the only relevant Theorem in the paper) from an underlying work and used it to motivate the analysis in the paper more directly.
Additionally, I notice that I was using both d and m for model parameter counts in the paper, I standardized on m for clarity.
Also, I had a language specialist proofread the paper to help clear up any lingering textual issues.
Round 2
Reviewer 2 Report (Previous Reviewer 1)
I think the manuscript is suitable for publication now.
This manuscript is a resubmission of an earlier submission. The following is a list of the peer review reports and author responses from that submission.
Round 1
Reviewer 1 Report
This paper applies a method called Information Corrected Estimation (ICE) to mortgage modeling and neural network training.
ICE is motivated from Takeuchi Information Criterion (TIC) and developed as a computationally stable version of TIC.
Some techniques for improving computational stability are also proposed.
The motivation of ICE is to improve computational instability of TIC, and I agree to the importance of this. However, theoretical validity of ICE is not clear. ICE has the same form with TIC, which is the sum of the log-likelihood and penalty term (trace of I J^{-1}). In the original derivation of TIC, this penalty term is obtained from the assumption that the parameter estimate maximizes log-likelihood (MLE) and Taylor expansion around it. Then, ICE uses the same form of penalty term for arbitrary value of the parameter. How is it justified? It would be nice if it has some natural interpretation, but I could not find it in the paper.
Also, I think the paper is not well organized enough and hard to understand in the current form.
- The first paragraph of introduction says "The ICE methodology is described in detail in [3], TIC is described in [9]. This section contains a very brief review to introduce some common notation needed for the 16 topics of this paper." I think it is too brief, because ICE is not well known to general audience.
- Section 2 is rather short and its relation to other sections is not clear.
- Some references are not complete, such as 3 and 5.
- Some bootstrap variants of information criterion have been proposed in the papers below. They are considered to improve the computational instability of TIC. So it would be nice to compare ICE with these methods.
Ishiguro, Sakamoto and Kitagawa. Bootstrapping Log Likelihood and EIC, an Extension of AIC. AISM 1997.
Konishi and Kitagawa. Bias and variance reduction techniques for bootstrap information criteria. AISM 2010.
Author Response
Thanks for the review. The references you provided will be helpful.
Reviewer 2 Report
The author studies TIC and ICE, which the author previously proposed. One of the problems with TIC for practical applications is that its calculation gives numerical instability. The author is trying to overcome the problem by considering ICE. The results are shown for several problems, and the technique developed by the author is expected to be applied to a broad class of problems. I understand what you want to do, but there are problems with the way it is written as a paper. In particular, the first section is only a review, and it is difficult to understand your motivation from the first section. I guessed your motivations and contributions from the abstract, but it would be better to explain them in the main text properReviewer 3 Report
This draft discusses the accuracy, stability and numerical cost of evaluating information criteria for model selection and model fitting. It is a follow-up of a recent paper introducing ICE, an extension of the wide-spread criterion TIC (Takeuchi Information Criterion). In the present draft, several computationally efficient approximations are discussed and evaluated on a large real world problem, a mortgage model. A detailed discussion shows how to efficiently implement these criteria within a neural network formulation using back-propagation. The proposed approximations and regularization of the expected Hessian matrix are shown to be numerically efficient while leading to good results compared with a vanilla TIC-based approach.
The presentation of the methodology is thorough and rather well written. The methodology is rigorously presented and the proposed approximations are carefully discussed. A mortgage problem is then considered to demonstrate the proposed approach. While the results show the benefit of using the regularized TIC and ICE, the presentation of this part is rather poor and contrasts with the clarity of the methodology section. This must be significantly reworked in order for the paper to be acceptable for publication, possibly following the comments below.
- The paper starts directly discussing the methodology without providing a context nor a discussion, even short, on the recent related efforts from the literature. A proper Introduction is required for a reasonably self-contained article.
- After Eq. (11), what is the meaning of "time" and "space" in "The computation of this term requires O(d3) time and O(d2) space"? Is "time" the number of floating point operations? And "space" the maximum memory requirement for storing temporary quantities?
- Around Eq. (16), it is said that since v(theta,x) is being reduced to zero as the optimization progresses, the first correction term in Eq. (15) should be larger than the second. Since this situation only is true at the optimum, or at a local extremum, it is unclear how to justify such an approximation in the general workflow. It might be less confusing to say that the second correction term is simply neglected, which is fully justified once the optimization has made progress. Otherwise, the magnitude of this term should be specifically monitored along the optimization process and shown to be negligible wrt the first correction term.
- The result section is unclear and lengthy. It needs to be significantly reworked. It would strongly benefit from being more concise and focusing the discussion and results in a synthetic manner. Too many situations are discussed with different combinations of approximations (objective function from Eq. (10) or (12), with stabilization (19) or not, with gradient from (17) or (18), etc. This must be reworked in a more synthetic way. Some tables may be advantageously replaced by plots, e.g. Tables 16, 17, 18. Please also specify the explicit expression of the variables considered instead of just a description; for instance, the quantity reported in Table 4.
- There are several typos which must be corrected. For instance, "[...] by a Akaike", "[...] it is shown the the [...]", "In Table reftab:adjCrossVal [...]", "For each regressor, for each quantization from 3 to 10 buckets [...]", "[...] this approach is given by [?], Section 4.1", "The approximation chosen here is ... ??" (missing end of sentence, section 1.5).